# Effects of Elevated Progesterone Levels on the Day of hCG on the Quality of Oocyte and Embryo

**DOI:** 10.3390/jcm11154319

**Published:** 2022-07-25

**Authors:** Jeesoo Woo, Hwang Kwon, Donghee Choi, Chan Park, Jihyang Kim, Jieun Shin, Jeehyun Kim, Youn-Jung Kang, Hwaseon Koo

**Affiliations:** 1Research Competency Milestones Program of School of Medicine, CHA University, 335 Pangyo-ro, Bundang-gu, Gyeonggi-do, Seongnam-si 13488, Korea; wooo@chauniv.ac.kr; 2Department of Obstetrics and Gynecology, CHA Fertility Center Bundang, 335 Pangyo-ro, Bundang-gu, Gyeonggi-do, Seongnam-si 13488, Korea; hkwon@chamc.co.kr (H.K.); dhchoimc@chamc.co.kr (D.C.); ifchan424@cha.ac.kr (C.P.); bin0902@chamc.co.kr (J.K.); 1219annie@cha.ac.kr (J.S.); jeehyun678@cha.ac.kr (J.K.); 3Department of Biochemistry, Research Institute for Basic Medical Science, School of Medicine, CHA University, 335 Pangyo-ro, Bundang-gu, Gyeonggi-do, Seongnam-si 13488, Korea

**Keywords:** progesterone levels, oocyte maturation, embryo development, fertilization, in vitro fertilization

## Abstract

This study is designed to investigate the effects of increased progesterone (P4) levels on the quality of retrieved oocytes and embryos during IVF. This retrospective analysis included 982 all-freezing in vitro fertilization (IVF) cycles (conducted between November 2019 and June 2020 at CHA Fertility Center Bundang, South Korea) in which serum P4 levels were measured on the day of human chorionic gonadotropin (hCG) administration. Our study revealed that the serum P4 levels on the day of hCG administration are strongly associated with the rates of oocyte maturation, displaying a positive correlation in patients with serum P4 < 2.25 ng/mL (*p* = 0.025). Moreover, patients with serum P4 < 1.25 ng/mL showed relatively low fertilization rates (*p* = 0.037), and the rates of good embryo retrieval were significantly increased with the serum P4 level < 1.5 ng/mL (*p* = 0.001). Interestingly, serum P4 level on the day of hCG administration affects the rate of good-quality embryo development, especially at the cleavage stage, and is associated with the status of ovarian responses. Our current study suggests that serum P4 level on the day of hCG administration negatively affects the rates of oocyte maturation, fertilization, and the development of good embryos.

## 1. Introduction

Progesterone (P4) elevation, on the day of human chorionic gonadotropin (hCG) administration, refers to rising progesterone levels in the absence of either premature luteinization (PL) or a luteinizing hormone (LH) surge during in vitro fertilization (IVF) cycles [1]. The prematurely elevated progesterone levels during the late follicular phase have the capacity to reduce the potential for pregnancy through IVF cycles by impairing endometrial receptivity [2,3]. Additionally, PL might induce the failure of the essential synchrony between the endometrium and the developing embryos [4]. However, the impact of increased serum progesterone levels on the quality of the embryos or oocytes is still debatable. For the first time, in 1993, an analysis was reported on the correlation between elevated progesterone levels and the quality of embryos [5]. Several reports have proposed that PL has little effect on the quality of embryos [3,6,7,8]. Additionally, an adverse effect of PL does not seem to be present in freeze–thaw (the ‘freeze-all’ strategy) and donor/recipient cycles [9]. However, Huang et al. [7] demonstrated that the rates of top-quality blastocyst formation are negatively correlated to P4 levels on the day of oocyte maturation in gonadotropin-releasing hormone (GnRH) antagonist cycles. Likewise, several studies have reported that elevated levels of progesterone have an adverse impact on embryo utilization rates, which were calculated as the total number of embryos transferred and cryopreserved divided by the total number of fertilized oocytes [10]. Progesterone receptors on the oocytes and embryos are widely known to have many functions that are strongly associated with oocyte meiosis and embryo development [11]. It is suggested that elevated progesterone levels affect oocyte and embryo production, regardless of evidence that elevated progesterone levels lower pregnancy rates by suppressing endometrial receptivity. However, with respect to the effect of elevated progesterone on endometrial receptivity, there is still minimal evidence as to whether the existence of elevated levels of progesterone influences the quality of oocyte and embryos. Here, we investigate the correlation between the levels of serum progesterone and the quality of embryos and oocytes to provide a therapeutic strategy to increase the rates of oocyte maturation and embryo development during IVF.

## 2. Materials and Methods

### 2.1. Study Design and Patients

To understand the mutual relationship between progesterone levels and the quality of embryos and oocytes, we used retrospective analyses of 982 all-freezing IVF cycles in which serum P4 levels were measured on the day of hCG administration. All data of patients were collected from the CHA Fertility Center, Bundang, between November 2019 and June 2020. The requirements for patient consent were waived due to the retrospective nature of the study. Moreover, the study design was approved by the Institutional Review Board of our hospital (IRB number: 2020-10-019-001). In this study, all patients received consistent IVF clinical treatments at the CHA Fertility Center, Bundang, and no additional intervention was performed. In addition, since this study only focused on the effects of elevated P4 levels on the rates of oocyte maturation and embryo development, infertility patients with factors, including the endometrium, fallopian tube, and male factors, were all included in the study group. The patients who failed to retrieve oocytes and took any supplements that might have affected the quality of oocytes and embryos (ex. growth hormone, antioxidant, etc.) were excluded. 

### 2.2. Protocol for Ovarian Stimulation and Hormone Measurements

Daily injection of FSH/LH mixed protocol under GnRH antagonist pituitary suppression was used for ovarian hyper-stimulation, which is usually started on the second or third day of the menstrual cycle. Gonadotropin doses were adjusted based on the status of ovarian response, and the patient’s weight and age, etc. A daily GnRH antagonist dose of 0.25 mg (Cetrotide, Merck Serono S.A., Geneva, Switzerland) was initiated, depending on the average size of follicles (≥12 mm in diameter), and was continued until hCG administration. When at least three follicles measured ≥18 mm in diameter, 5000–10,000 IU of recombinant hCG (Ovidrel, Merck Serono S.A., Rockland, MA, USA) was administered. Serum progesterone and estradiol levels were measured on the day of hCG administration. Oocytes were retrieved 34–36 h after hCG administration. We used both conventional IVF and intracytoplasmic sperm injection (ICSI) for the process of fertilization, as clinically indicated.

### 2.3. Embryo Culture and Grading

Briefly, semen was collected in a sterile container through masturbation after 3–5 days of sexual abstinence and maintained at 37 °C for 30 min. After liquefaction, samples were analyzed for sperm concentration, motility and morphology, according to the World Health Organization criteria. The oocytes were incubated in G-IVF medium (Vitrolife, San Diego, CA, USA) and fertilization occurred 3 to 4 h after retrieval. We confirmed that mature oocytes were defined as a structure known as the polar body. Normal fertilization was defined as zygotes with two pronuclei (2PN). Then, fertilized oocytes were continuously cultured in a G1 medium for 2 more days. Day 3 embryos were evaluated on the basis of the numbers and symmetry of their blastomeres, the percentage of fragmentation, vacuolization, granulation and multi-nucleation [10]. The grades for the blastocysts were evaluated according to the Gardner and Schoolcraft grading system based on the expansion stage, the number of cells joining compaction or blastulation, and the appearance of the trophectoderm and inner cell mass (Table 1).

### 2.4. Data Collection

Data were obtained from computerized databases. Patient information, including age, BMI, levels of FSH, GnRH, peak serum E2, LH, and P4, on the day of hCG administration, and the status of freezing follicles and embryos, were evaluated. 

### 2.5. Statistical Analysis

All data were sorted using Python 3.8. An independent t-test was used to compare the basal characteristics, and the categorical variables, including the rates of pregnancy and clinical pregnancy, were compared using the chi-square test or the Fisher exact test. The Pearson correlation coefficient was calculated to estimate the correlations between pregnancy and multiple variables, using the Statistical Package for the Social Sciences for Windows (SPSS, Chicago, IL, USA). A *p*-value of < 0.05 was considered statistically significant. Patients were divided into 11 distinct groups, according to their serum P4 levels on the day of hCG administration: <0.25, 0.25–0.5, 0.5–0.75, 0.75–1, 1–1.25, 1.25–1.5, 1.5–1.75, 1.75–2.0, 2.0–2.25, 2.25–2.5, ≥2.5 ng/mL. These cut-off levels were chosen to provide equal intervals focused on the threshold. These threshold values provide meaningful results after several trials by referring to previous studies [12]. 

## 3. Results

### 3.1. Basic Characteristics

A total of 809 patients, who underwent 1006 IVF cycles, were evaluated. The average age of the participants was 37.2 ± 4.4 years, the average BMI was 21.9 ± 3.2, and the average number of oocytes retrieved was 10.9 ± 8.7. The clinical features and cycle outcomes of enrolled cycles are summarized in Table 2.

### 3.2. The Correlation between the Levels of Serum Progesterone and the Rates of Oocyte Maturation, Fertilization, and Good-Quality Embryo

We evaluated the effects of serum P4 levels on the day of hCG administration on the quality of oocytes and embryos. The rate of oocyte maturation was defined as the ratio of the number of mature oocytes with the polar body out of the total oocytes retrieved. The definitions of the clinical parameters used in this study are the following: Maturation rate = number of MII oocytes/number of total retrieved oocytes; fertilization rate = number of 2PN/number of attempts of fertilization; good-quality embryo (GQE) rate of cleavage stage (CL) or BL = number of GQE of CL or BL/ total number of CL or BL (Table 1). The overall association between the levels of serum P4 on the day of hCG administration, and the rates of oocyte maturation, fertilization, and GQE production, were analyzed. The linear regression analyses demonstrate no significant association between the rates of maturation, fertilization, the GQE of CL, and serum P4 level (Figure 1).

### 3.3. The Effects of Serum Progesterone Levels on the Rate of Mature Oocyte

Our analyses revealed that the rates of oocyte maturation were negatively correlated with serum P4 levels on the day of hCG administration. The maturation rates were significantly higher in patients whose serum P4 levels were maintained lower than 2.25 ng/mL compared with those with serum P4 levels > 2.25 ng/mL (*p* = 0.025) (Figure 2). 

### 3.4. The Effect of Serum Progesterone Levels on the Rate of Fertilization 

We evaluated the correlation between the levels of serum P4 on the day of hCG administration and the rates of fertilization. Interestingly, patients with serum P4 level ≥ 1.25 ng/mL showed noticeably lower rates of fertilization than those with P4 levels < 1.25 ng/mL (*p* = 0.037) (Figure 3).

### 3.5. The Effect of Serum Progesterone Levels on the Rate of Good-Quality Embryo (GQE) 

Patients with serum P4 level ≥1.75 ng/mL on the day of hCG administration showed a negative association with GQE rates at the cleavage stage (CL) (Figure 4A). However, there was no significant association observed between serum P4 levels and the rates of GQE at the blastocyst stage (BL) (Figure 4B). 

### 3.6. The Effect of Serum Progesterone Levels on the Various Parameters According to Patient’s Age 

We next evaluated the effect of serum P4 levels on the rates of oocyte maturation, fertilization, and embryo development, depending on the age of patient. Table 3 shows the correlation analyses of the rates of oocyte maturation, fertilization, and GQE development within the association of P4, P4/oocyte, or P4/E2. P4/oocyte denotes the serum P4 level on the day of hCG administration divided by the number of oocytes retrieved. Moreover, P4/E2 indicates the serum P4 level divided by the peak E2 level on the day of hCG administration. Intriguingly, it was observed that the rates of GQE development at the cleavage stage were significantly decreased as the serum P4 levels increased in the 40–45-year-old patient group (*p* = 0.038). 

### 3.7. The Effect of Serum Progesterone Levels on the Various Parameters According to the Number of Retrieved Oocytes 

We analyzed the effect of serum P4 levels on the quality of oocytes or embryos within the association of the numbers of oocytes retrieved. Patients were categorized into three groups based on the status of ovarian response: low ovarian responder (≤3 oocytes retrieved), sub-optimal (4–6 oocyte retrieved), normal (7–12 oocyte retrieved) and high ovarian responder (≥13 oocytes retrieved). Interestingly, the rates of GQE development at the cleavage stage were significantly associated with serum P4 levels in the groups of poor ovarian responders; however, there was no significant association observed in the intermediate ovarian responders (Table 4).

### 3.8. The Effect of Serum Progesterone Levels on Pregnancy Rates and Clinical Pregnancy Rates 

We finally analyzed the effects of serum P4 levels on the rates of pregnancy or clinical pregnancy in both CL and BL embryo transfer cycles. Among 1006 IVF cycles, 661 frozen embryo transfer cycles were performed (CL embryo transfer: 426 cases, BL embryo transfer: 235 cases). Each frozen embryo transfer cycle was carried out immediately after the collection of good embryos, through several cycles, and further proceeded with natural or artificial cycles. The β-hCG test was performed after progesterone administration or 14 days after ovulation, and β-hCG >10 mIU/mL was defined as pregnancy and clinical pregnancy rate, recorded when the gestational sac was confirmed around 5–6 weeks of amenorrhea. Our analyses revealed that no correlations were found between serum P4 levels and the rates of pregnancy (Figure 5A,B) or clinical pregnancy (Figure 6A,B) in both cleavage and blastocyst stage embryo transfer cycles, even though several statistically significant cut-off levels were observed in the cleavage stage transplantation cycle. 

## 4. Discussion

This retrospective study, including 1006 fresh IVF cycles, demonstrates that the elevated levels of serum P4, on the day of hCG administration, showed a negative effect on the quality of retrieved oocytes and embryos. It has been widely reported that elevated P4 levels are associated with degraded endometrial receptivity [13,14,15,16]. However, there are limited data available on the impact of P4 levels on the quality of oocytes and embryos. In this study, levels of P4, on the day of hCG administration, have been shown to adversely affect the rates of oocyte maturation, fertilization, and GQE development during IVF cycles. In particular, when serum P4 levels were > 1.75 ng/mL, the rates of GQE at the cleavage stage were significantly reduced. Some analyses have indicated that there is a significant association with a decreased probability of pregnancy during IVF cycles with detrimental endometrial receptivity under a serum P4 threshold > 1.5 ng/mL [17,18,19]. However, the clinical criteria for increased P4 levels are highly dependent on individual clinics [9]. Therefore, these levels are yet to be conclusive. Moreover, rarely are there studies presenting the threshold of P4 levels that might affect the quality of oocytes or embryos during IVF. The intervals of the threshold of P4 levels used in our analyses were randomly selected; each threshold with 0.25 ng/mL of intervals ranging from 0.25 ng/mL to 2.25 ng/mL. As a result, oocyte maturation rates were decreased at P4 ≥ 2.25 ng/mL, fertilization rates were also decreased at P4 ≥ 1.25 ng/mL, and the ratio of GQE at the cleavage stage was decreased at P4 ≥ 1.75 ng/mL. Although it is not clear exactly how P4 affects the quality of oocytes and embryos, several animal studies have previously shown that P4 plays a key role in oocyte formation and maturation, fertilization, and embryo implantation and development [11,20,21,22]. Three isoforms of progesterone receptors (PR) exist in the follicle, and the physiological effects of P4 are mediated through PR [23,24]. PR-A and PR-B, two specific isoforms of nuclear PRs, are expressed in pre-ovulatory follicle granulosa cells [25,26]. It is remarkable that membrane PR or non-genomic PGR acts as a promoter of oocyte meiosis. Aparicio et al., have shown that intracellular signaling of P4 is achieved by its interaction with nuclear and membrane PRs and is involved in oocyte developmental competence, implicating its critical role in the formation of mature oocytes [27,28]. Even though there are several controversial reports on the impact of P4 [29,30,31], there is no study that investigated a cut-off concentration of P4 that negatively affects oocyte quality and fertilization rates. According to our findings, it is essential that an efficient strategy is established to improve the quality of oocytes and fertilization rates during IVF cycles. 

Studies in which PL negatively affects the quality of embryos are often reported; however, there is a matter of continuing debate. It has been reported that the ratio of GQE is significantly decreased with a P4 level of 2.0–2.5 ng/mL [7]. In our findings, it was confirmed that the GQE rate at the cleavage stage was statistically decreased when serum P4 levels were > 1.75 ng/mL (*p* = 0.030). Although the cut-off level is different from the study by Huang et al., similar results were obtained [7]. In addition, in our study, there were no significant differences in the GQE rate of the blastocyst stage between serum P4 levels below the and above the cut-off points. (Figure 4B). However, there was a study to investigate the top-quality blastocyst formation rate within the relation of P4 levels during IVF/ICSI GnRH antagonist cycles. This was the first study to establish that a detrimental effect of elevated P4 levels on the quality of day 5-6 embryo was observed with regard to the rates of top-quality embryo formation [32]. The exact reasons for these results are still controversial. However, according to several studies, PRs exist on the surface of the embryo before the blastocyst stage, although they disappear thereafter. Based on this, PL might affect the quality of embryos at the cleavage stage, but not at a later stage, such as the blastocyst stage. However, further research is required for a detailed explanation. Chingwen Ying et al., reported that PR mRNA and protein were expressed during the pre-implantation stages of pig embryos until the 4-cell stage, but not at the 8-cell to later stages through blastocyst formation [33]. The presence of PR mRNA was not detected prior to the blastocyst stage. The blastocyst stage deserves further research based on previous results.

Yan-Guang Wu et al., demonstrated that the frequency of PL increases as the ovarian function declines with women’s increased age [34]. In our findings, the prematurely elevated serum P4 levels did not affect the quality of oocytes and embryos in patients <40 years. However, our findings revealed that prematurely elevated serum P4 levels showed a negative effect on the fertilization rate and quality of cleavage stage embryos in patients over 40 years old. Although it is not clear why the incidence of PL increases with the age of women, various studies with animals suggest that a possible answer might be the aberrant function of aged granulosa cells. Hence, age is the critical factor that should be considered when investigating the impact of prematurely elevated serum P4 levels on oocytes and embryos. It is also known that the number of collected oocytes is also related to the frequency of PL. Thus, several researchers recommended considering the ratio of P4 to E2, or P4 to the number of collected oocytes, as good predictors, rather than the use of the absolute value of P4 [35,36,37,38,39]. Correspondingly, our study evaluated the effect of P4/oocyte ratios on oocytes’ and embryos’ quality. It was confirmed that elevated P4, P4/oocyte and P4/E2 had a negative effect on the embryos at the cleavage stage in poor responders (less than three oocytes). This might imply that the poor-responder group is more sensitive to serum P4 with regard to the quality of oocytes and embryos than the normal-responder group (Table 4). In order to clearly understand the effect of P4 on the embryo, it is most accurate when it comes with the associated analyses of the pregnancy rates. However, due to the nature of strategies of freeze-all cycles, after the collection and selection of good embryos, which are mainly applied in the current institution, there is the disadvantage of having an exact match between transplanted embryos and the cycle from which the embryos are retrieved. In addition, pregnancy rates are affected by various factors, including the quality of the embryos. 

## 5. Conclusions

Our current study, in which 1006 fresh IVF cycles were analyzed, clearly demonstrates that the elevated levels of serum P4 on the day of hCG administration have negative effects on the rates of oocyte maturation, fertilization, and good embryo development (CL). These data suggest that maintaining serum P4, at least at less than 2.25 ng/mL, might be helpful for increasing the rates of good oocyte maturation (*p* = 0.025). Elevated P4 levels (>1.25 ng/mL) consistently seem to be detrimental to fertilization. The limitation of this study is that data were collected by retrospective analyses. However, the impact of PL on fertility is already largely known; our current study mostly focused on its effect on the quality of oocytes and embryos. Further studies are essential for understanding the effects of P4 levels on embryo quality and pregnancy rates when associated with other various factors that mediate the successful implantation and maintenance of pregnancy. Indeed, in our study, elevated serum P4 levels showed detrimental effects on oocyte maturation and the early stages of embryo development. However, no significant effect was observed in the prolonged culture to the blastocyst state, which strongly suggests a single embryo transfer at the blastocyst stage in a freeze-all cycle as an effective strategy to overcome P4 effects to increase the proportion of high-quality embryos during IVF-ET, resulting in improved pregnancy rates. However, there are many cases in which the freeze-all method is not applicable, and the transfer of a fresh embryo is the only remaining option. Based on our findings, we encourage maintaining an appropriate level of superovulation in cases of the repeated failure of good-quality embryo retrieval.

## Figures and Tables

**Figure 1 jcm-11-04319-f001:**
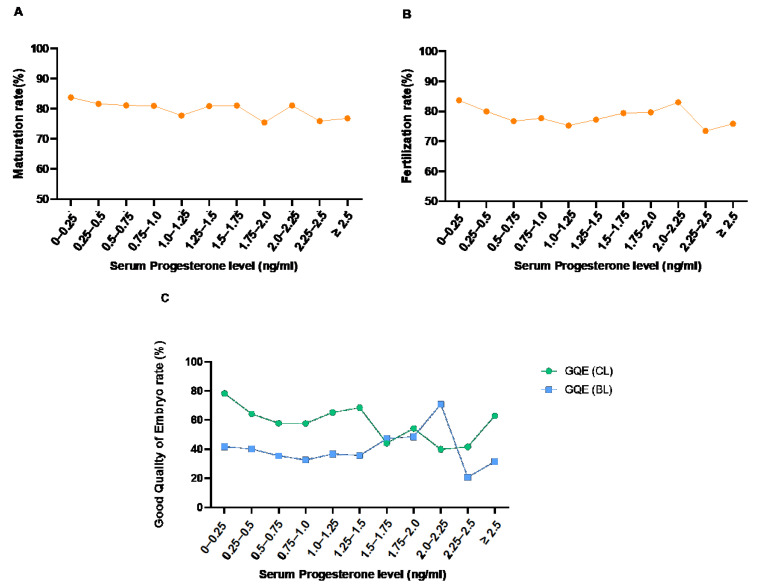
Association between the levels of serum progesterone on the day of hCG administration and maturation rate (**A**), fertilization rate (**B**), and good-quality embryo (GQE) rate (cleavage stage (CL) and blastocyst stage (BL)) (**C**).

**Figure 2 jcm-11-04319-f002:**
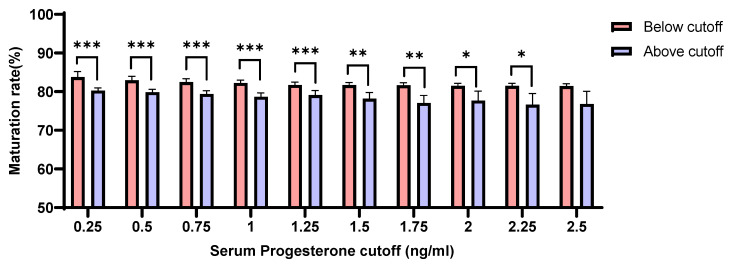
Association of the rates of oocyte maturation with the concentration of serum P4 levels (below or above each cut-off) on the day of hCG administration (*p* < 0.05 (*), *p* < 0.01 (**), *p* < 0.001 (***)).

**Figure 3 jcm-11-04319-f003:**
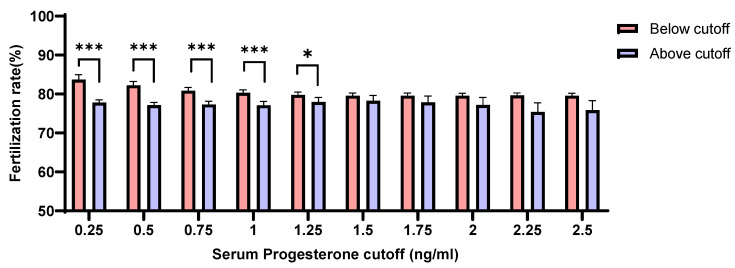
Association of fertilization rates with the concentration of serum progesterone (below or above each cut-off) on the day of hCG administration (*p* < 0.05 (*), *p* < 0.001 (***)).

**Figure 4 jcm-11-04319-f004:**
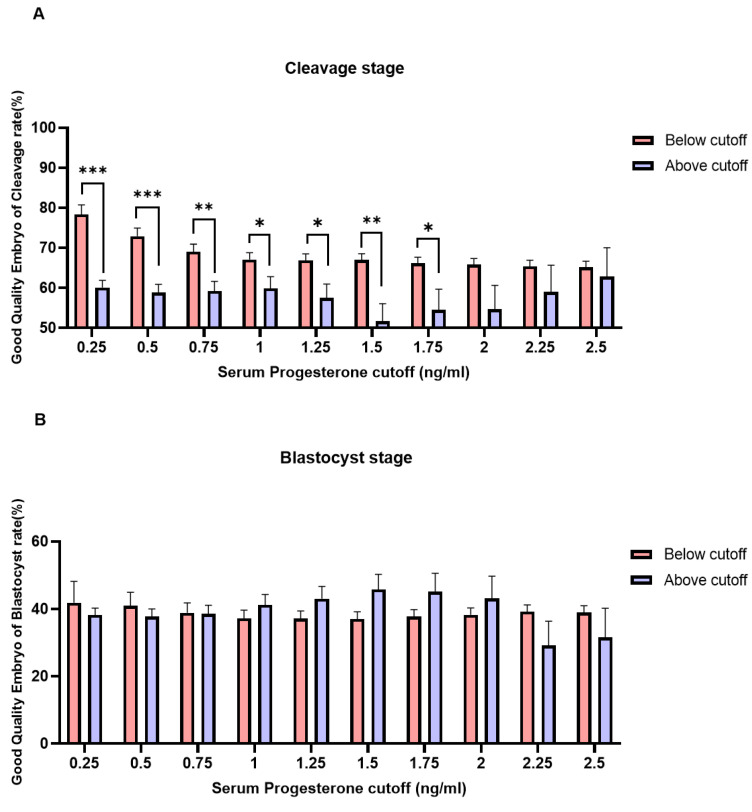
Correlation between serum P4 level on the day of hCG administration and the rates of good-quality embryo (GQE) at the cleavage stage (**A**) and the blastocyst stage (**B**) (*p* < 0.05 (*), *p* < 0.01 (**), *p* < 0.001 (***).

**Figure 5 jcm-11-04319-f005:**
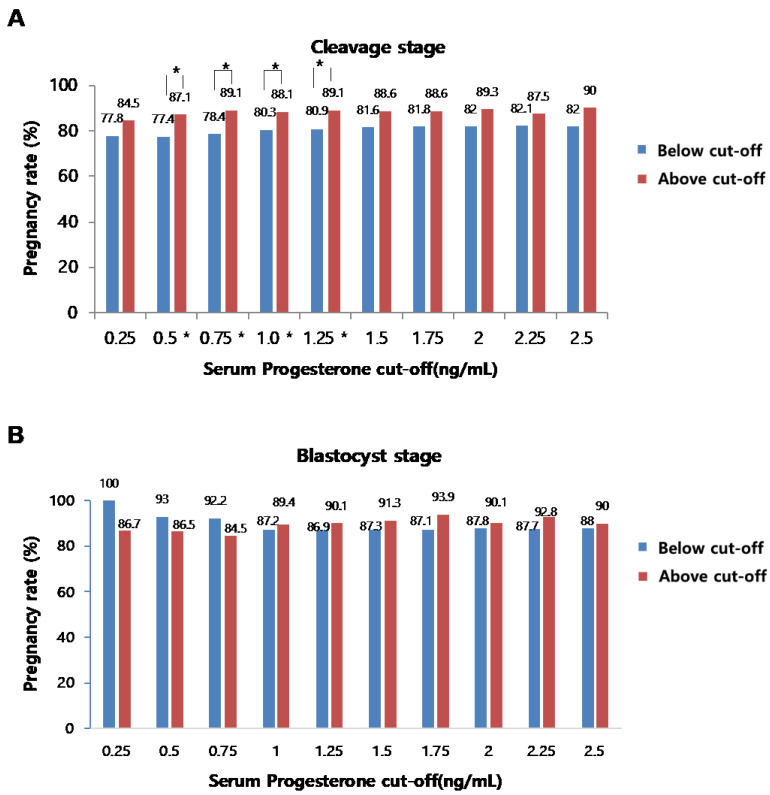
Correlation between serum P4 level on the day of hCG administration and the pregnancy rates at the cleavage stage (**A**) and blastocyst stage (**B**) (*p* < 0.05 (*)).

**Figure 6 jcm-11-04319-f006:**
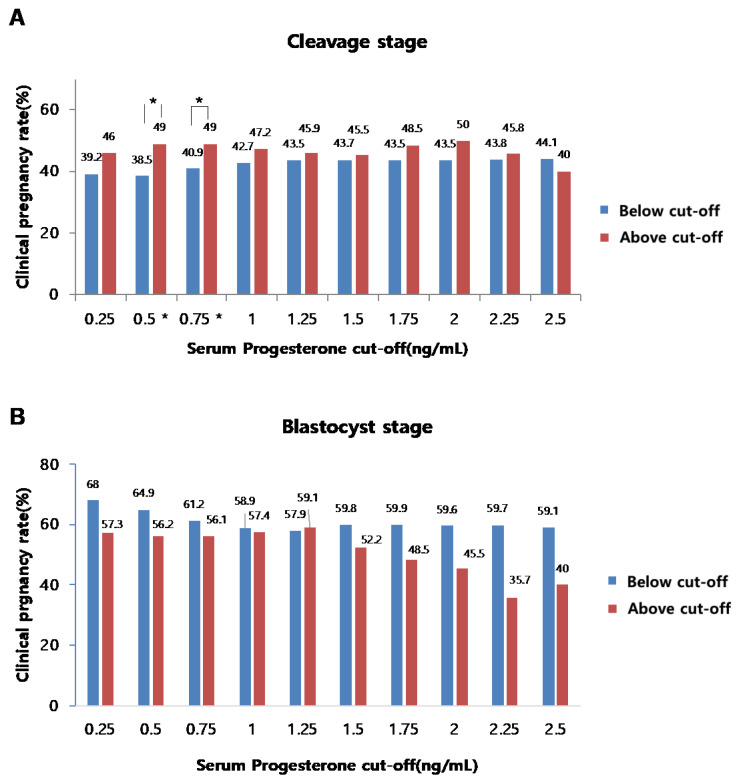
Correlation between serum P4 level on the day of hCG administration and the clinical pregnancy rates at the cleavage stage (**A**) and the blastocyst stage (**B**) (*p* < 0.05 (*)).

**Table 1 jcm-11-04319-t001:** The definition of a good-quality embryo (GQE).

	Excellent	Good	Bad
Cleavage-stage	G1, G2	G3	G4, G5, G6
Blastocyst-stage	4AA, 5AA, 6AA	3AA, 3AB, 3BA, 3BB, 4AB, 4BA, 4BB, 5AB, 5BA, 5BB, 6AB, 6BA, 6BB	1AB, 1BA, 1BB, 1CC, 2AA, 2AB, 2BA, 2BC, 2CB, 2CC, 3AC, 3BC, 3CB, 3CC, 4AC, 4BC, 4CB, 4CC, 5BC, 5CB, 5CC

**Table 2 jcm-11-04319-t002:** Basic characteristics of patients for the ovarian stimulation of IVF cycle.

Parameter	
Age (y)	37.15 ± 4.38
BMI (kg/m^2^)	21.86 ± 3.21
Basal FSH (IU/L)	11.09 ± 9.24
Basal AMH (ng/mL)	2.27 ± 2.34
No. oocyte retrieved	10.92 ±8.72
Fertilization rate (%)	79.42 ± 18.89
Peak estradiol (pg/mL)	2246.00 ± 2034.78
Progesterone on hCG day (ng/mL)	0.86 ± 1.33

**Table 3 jcm-11-04319-t003:** Correlation analysis of parameters related to P4, P4/oocyte and P4/E2 according to patient’s age.

No. of Oocytes Retrieved			≤35		35≤ <40		40≤ <45		45≤	
			(*n* = 253)		(*n* = 436)		(*n* = 263)		(*n* = 30)	
Parameter			Pearson C.C	*p* Value	Pearson C.C	*p* Value	Pearson C.C	*p* Value	Pearson C.C	*p* Value
Maturation rate		P4	0.006	NS	−0.052	NS	−0.090	NS	0.112	NS
		P4/oocyte	0.061	NS	−0.083	NS	−0.062	NS	0.380	0.038 *
		P4/E2	0.045	NS	−0.017	NS	−0.077	NS	0.085	NS
Fertilization rate		P4	−0.002	NS	−0.018	NS	−0.132	0.032 *	0.116	NS
		P4/oocyte	0.090	NS	0.042	NS	−0.031	NS	0.335	NS
		P4/E2	0.060	NS	0.014	NS	−0.104	NS	0.157	NS
GQE	CL	P4	−0.018	NS	−0.091	NS	−0.184	0.003 **	0.424	0.020 *
		P4/oocyte	0.071	NS	0.031	NS	−0.125	0.042 *	0.003	NS
		P4/E2	−0.064	NS	0.068	NS	−0.120	NS	−0.052	NS
	BL	P4	0.000	NS	0.042	NS	0.042	NS	0.791	0.000 **
		P4/oocyte	−0.095	NS	−0.087	NS	−0.049	NS	0.002	NS
		P4/E2	0.035	NS	−0.083	NS	−0.049	NS	−0.068	NS

GQE, good-quality embryo; P4, progesterone; E2, estradiol; Pearson C.C., Pearson correlation coefficients; NS = not significant; **, Correlation is significant <0.01 level.; *, Correlation is significant <0.05 level.

**Table 4 jcm-11-04319-t004:** Correlation analysis of parameters related to P4, P4/oocyte and P4/E2 according to the number of retrieved oocytes.

No. of Oocytes Retrieved			Low: 0−3		Sub-Optimal: 4−6		Normal: 7−12		High: ≥13	
			(*n* = 236)		(*n* = 188)		(*n* = 263)		(*n* = 295)	
Parameter			**Pearson C.C**	***p* Value**	**Pearson C.C**	***p* Value**	**Pearson C.C**	***p* Value**	**Pearson C.C**	***p* Value**
Maturation rate		P4	−0.148	NS	0.036	NS	−0.131	0.030 *	0.000	NS
		P4/oocyte	−0.184	0.005 **	−0.015	NS	−0.081	NS	−0.018	NS
		P4/E2	−0.088	NS	−0.025	NS	−0.091	NS	0.001	NS
Fertilization rate		P4	−0.185	0.029 *	0.087	NS	−0.005	NS	0.002	NS
		P4/oocyte	−0.073	NS	0.083	NS	0.005	NS	−0.001	NS
		P4/E2	−0.104	NS	0.028	NS	0.009	NS	0.038	NS
GQE	CL	P4	−0.348	0.000 **	−0.192	0.021	0.047	NS	−0.024	NS
		P4/oocyte	−0.321	0.000 **	−0.104	NS	0.066	NS	−0.018	NS
		P4/E2	−0.265	0.000 **	−0.089	NS	0.070	NS	−0.021	NS
	BL	P4	−0.154	NS	−0.136	NS	−0.040	NS	−0.004	NS
		P4/oocyte	−0.130	NS	−0.099	NS	−0.028	NS	0.009	NS
		P4/E2	−0.084	NS	−0.108	NS	−0.048	NS	−0.024	NS

GQE, good-quality embryo; P4, progesterone; E2, estradiol; Pearson C.C., Pearson correlation coefficients; NS = not significant; **, Correlation is significant <0.01 level.; *, Correlation is significant <0.05 level.

## Data Availability

The datasets used and/or analyzed during the current study are available from the corresponding authors on reasonable request.

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
