# Peer review of "Effects of Elevated Progesterone Levels on the Day of hCG on the Quality of Oocyte and Embryo"

_jcm, 2022, doi:10.3390/jcm11154319_

Round 1

Reviewer 1 Report

The presented paper seems to be too preliminary. Additional data, statistical analysis are mandatory. Describing the results in the tables and figures are not self-explanatory. The number of references are not enough.

Reviewer 2 Report

Well written paper on a topic that remains a matter of debate. However, sound scientific work with clear results. Should be transmitted to the scientific community. Implantation rates, as well as clinical pregnancy rates and baby take home rates after FET should be added and discussed under the pespective of the impact of P4 elevation on final clinical results. 

Reviewer 3 Report

I read with interest the current paper regarding the effect of increased P4 levels on the quality of retrieved oocytes and embryos during IVF. The manuscript is clear and well writte, however some issues should be addressed before publication.

- I would suggest to reconsider the correlation analysis according to the number of retrieved oocytes and clustering the patients as follows: poor responders (≤3 oocytes), sub-optimal (4-6), normal (7-12), high (≥13).

- The main finding of the manuscript is represented by the detrimental effect of P4 increase in the early stages of oocyte maturation and embryo development. On the contrary, the prolonged culture to the blastocyst stage seems to not be affected. Single embryo transfer at the blastocy stage in a freeze-all cycle (cycle segmentation) represents the ideal strategy to overcome P4 effect. Please further reinforce this concept in discussion

- The main limit of the study is the lack of any clinical outcomes. Please clarify that no robust conclusion can be drawn unless a reduced clinical pregnancy and live birth rates are shown.

Round 2

Reviewer 1 Report

The paper has not been improved enough. Well, I have some major comments:

1. Data Availability Statement should be corrected because it is a research article. 

2. Tables, references and whole manuscript have to be prepared according to the journal's requirements. 

3. Some typo mistakes (for example line 115; figure 1 - ng/ml) need to be eliminated.

4. The title should be shortened and more informative. 

5. When the authors used non-parametric statistical methods (Kruskal-Wallis test, U Mann Withney test) all results (in the text, tables and figures) have to be shown as Median, down quartile, upper quartile. It has to be corrected. ME, Q1, Q3 and interquartile range have to shown in the figure. In the current form are not acceptable.

6. in discussion section, please add strengths and limitations, future perspectives of this study.

7. Please add and explain more clinical significance from obtained results.

Reviewer 3 Report

All my comments have been addressed

Author Response

Thank you very much for your insightful and critical comments.